# Clinical Developmental Cardiology for Understanding Etiology of Congenital Heart Disease

**DOI:** 10.3390/jcm11092381

**Published:** 2022-04-24

**Authors:** Hiroyuki Yamagishi

**Affiliations:** Department of Pediatrics, Keio University School of Medicine, Tokyo 160-8582, Japan; hyamag@keio.jp

**Keywords:** left–right axis, heterotaxy, Fontan, outflow tract, neural crest, second heart field

## Abstract

Congenital heart diseases (CHD) result from abnormal development of the cardiovascular system and usually involve defects in specific steps or structural components of the developing heart and vessels. The determination of left–right patterning of our body proceeds by the steps involving the leftward “nodal flow” by motile cilia in the node and molecules that are expressed only on the left side of the embryo, eventually activating the molecular pathway for the left-side specific morphogenesis. Disruption of any of these steps may result in left–right patterning defects or heterotaxy syndrome. As for the outflow tract development, neural crest cells migrate into the cardiac outflow tract and contribute to form the septum of the outflow tract that divides the embryonic single truncus arteriosus into the aortic and the pulmonary trunk. Reciprocal signaling between neural crest cells and another population of myocardial precursor cells originated from the second heart field are essential for the steps of outflow tract development. To better understand the etiology of CHD, it is important to consider what kind of CHD is caused by abnormalities in each step during the complex development of the cardiovascular system.

## 1. Introduction

### 1.1. Region-Specific Step-by-Step Understanding of Cardiovascular Development for Congenital Heart Disease

Cardiovascular development in higher vertebrates involves a number of complex processes that are temporally and spatially orchestrated: migration, proliferation, differentiation, programmed cell death, and interaction of cardiac progenitor cells of different origins [1,2]. In order to better understand this complex process, it may be helpful to divide it into several regions or steps [3] (Figure 1). At each step, we should understand how each region of the cardiovascular system is formed by which cellular and molecular mechanisms, so that we can grasp the whole picture. As a matter of fact, most congenital heart diseases that we encounter in our daily practice are specific developmental abnormalities in one of these regions, while general developmental abnormalities lead to embryonic lethality. Therefore, this concept is also important for understanding the etiology of congenital heart disease. In other words, it is important to consider what kind of congenital heart disease is caused by abnormalities in each region during the development of the complex cardiovascular system [3].

### 1.2. Early Step of Cardiovascular Development and Heterotaxy Syndrome

The early embryonic heart is a single tubular structure (primitive heart tube) formed in the midline of the embryo, and at this stage the entire embryo has a symmetrical morphology. The rightward looping of the primitive heart tube is the first developmental stage in which asymmetry of internal organs appears in the embryo. In order for the organs and tissues of the body to form normally and asymmetrically, information about the left–right axis of the body is necessary [1,2,3,4]. When this information is disturbed, heterotaxy syndrome develops, which is associated with serious congenital heart diseases such as single ventricle.

### 1.3. Development of Left–Right Axis

The process of determining the left–right axis consists of the following four steps: (1) rotational movement of cilia in the node, (2) nodal flow: right-to-left flow of the embryo, (3) expression of the “left-sided formation mechanism” on the left side of the embryo, and (4) activation of left-sided formation molecules and genes (Nodal–Lefty–Pitx2) [4,5] (Figure 2).

The cells of the primitive node formed in the early embryo have cilia that are composed of microtubules with motor protein and dynein, existing between the microtubules. The “helical” motion of microtubules causes the cilia to rotate in a certain direction [4,5]. As the axis of rotation is at a certain oblique angle, it results in a nodal flow from the right to the left side of the embryo which carries some morphogenetic factors essential for the determination of the left–right axis to the left side of the embryo. Nodal, a ligand protein belonging to the TGF-β family, is initially expressed evenly around the node but diffuses to the left side of the node by “nodal flow”. Nodal phosphorylates and activates downstream SMAD factors through binding to its receptor, ActRII [6]. The expression pattern of Nodal in the lateral plate mesoderm is restricted to the left side, with no expression on the right side. Lefty1/2, such as Nodal, is a member of the TGF-β family and acts in a repressive manner against Nodal by competing with Nodal for binding to ActRII. While Nodal promotes self-expression through a positive feedback mechanism, it induces the expression of Lefty1 in the midline and Lefty2 in the left lateral plate mesoderm. Lefty1 prevents the rightward spread of Nodal and other left–right determinants, while Lefty2 regulates the expression of Nodal in the left lateral plate mesoderm. Lefty2 regulates the expression of Nodal in the left lateral plate mesoderm. Nodal signaling specific to the left lateral plate mesoderm region induces the downstream transcription factor Pitx2, which transmits information about the left–right axis to organ morphogenesis [6]. Pitx2 expressed in the left lateral plate mesoderm acts on precardiac mesoderm and is ultimately involved in determining the left side of the cardiac inflow and outflow tracts.

### 1.4. Developmental Defects of Left–Right Axis and Heterotaxy Syndrome

If the left-sided formation mechanism is not expressed in the left side of the embryo but in the right side, it may cause “visceral inversion”; if it is not expressed in both sides, it may cause “right isomerism” or asplenia syndrome, and if it is expressed in both sides, it may cause “left isomerism” or polysplenia syndrome. Heterotaxy is the result of randomization of the left–right differentiation information of each organ and tissue [6]. The causes of heterotaxy syndrome have been identified as (1) abnormalities in genes related to structural proteins of the lineage [6] and (2) abnormalities in genes related to the left lateral formation mechanism [6], which are involved in the development of the left–right axis as described above.

### 1.5. Morphological Characteristics of Heterotaxy Syndrome

Heterotaxy syndrome is a general term for diseases based on abnormalities in the position of thoracoabdominal organs in relation to the left–right axis of the body, encompassing asplenia and polysplenia syndromes [6]. The asplenia syndrome is characterized by right isomerism, in which the spleen, which should normally develop on the left side, is defective and the right organ develops symmetrically on the left side, while the polysplenia syndrome is characterized by left isomerism, in which the left organ develops symmetrically on the right side. However, spleen morphology may not always reflect left–right isomerism, and in individual cases, various degrees of impaired differentiation and mixed left–right isomerism (situs ambiguous) can be observed in the location of each organ, ranging from normal to isomerism.

### 1.6. Characteristics of Asplenia Syndrome and Necessary Medical Care

In the right isomerism characteristic of asplenia syndrome, a common atrioventricular canal, single ventricle, single atrium, abnormal pulmonary venous return, pulmonary artery obstruction/stenosis, and transposition of the great arteries may be associated in combination [6]. Cyanosis, due to right–left shunt and/or decreased pulmonary blood flow and/or heart failure associated with pulmonary congestion and/or atrioventricular valve regurgitation, may be seen from the neonatal period. The treatment strategy often aims at Fontan-type operations. In general, right isomerism heart disease is more complicated and has a worse prognosis than left isomerism heart disease, and the prognosis is influenced by (1) the presence of pulmonary artery obstruction or stenosis, (2) the presence of abnormal total pulmonary venous return and/or pulmonary vein obstruction, and (3) the degree of common atrioventricular valve regurgitation. In addition, there are cases in which two sinus and/or atrioventricular nodes, which normally develop on the right side, lead to paroxysmal supraventricular tachycardia [6].

The asplenia syndrome is a high-risk group for bacterial infections such as severe pneumococcal infections, as is the case after splenectomy. According to a report by the Japanese Society of Pediatric Cardiology Committee on the epidemiology of severe infections in Japan [7], the frequency of severe infections in asplenia syndrome was higher than that of bacterial endocarditis in high-risk congenital heart disease, with a mortality rate of 19%. *S. pneumoniae* and *H. influenzae* are particularly important as causative organisms. In Japan, simultaneous vaccination with sedimented 13-valent pneumococcal conjugate vaccine and Hib vaccine is available from 2 months of age, and 23-valent pneumococcal capsular polysaccharide vaccine is also applicable after 2 years of age. The prophylactic administration of antimicrobial agents may be recommended.

### 1.7. Characteristics of Polysplenia Syndrome and Necessary Medical Care

In the left isomerism syndrome characteristic of polysplenia syndrome, atrioventricular septal defect with biventricular atrioventricular connection and abnormal systemic venous return are common, and increased pulmonary blood flow and congestive heart failure are seen from early infancy. Although biventricular repair is possible in many cases compared with right isomerism cases, the prognosis is poor in cases of hypoplastic left heart and severe pulmonary hypertension [6]. We have reported that incomplete atrioventricular septal defect, which does not usually cause pulmonary hypertension in childhood, combined with polysplenia syndrome is an independent risk factor for the development of pulmonary hypertension, regardless of the age at the time of evaluation or the degree of left–right shunt [8]. The reason for this is speculative, but it may be that both sides of the lungs are left-sided, resulting in a smaller volume and fewer pulmonary vascular beds, and/or that the genetic abnormality that causes the polysplenia may also be involved in the development of pulmonary hypertension. In any case, incomplete atrioventricular septal defect associated with polysplenism requires early intervention to control pulmonary blood flow to avoid the development of pulmonary hypertension. In addition, the sinus node and atrioventricular conduction system, which normally develop on the right side, may be hypoplastic, resulting in bradyarrhythmia (sinus node disfunction and/or atrioventricular block) and, eventually, need for a pacemaker in some cases [6]. The inferior vena cava, which normally arises on the right side of the body, is absent in polysplenia and is often associated with congenital anomalies of the liver and biliary system, such as biliary atresia. In relation to congenital anomalies of the liver and biliary system, it is also necessary to be aware of congenital porto-systemic shunts (CPSS), a condition in which the portal vein forms a congenital shunt into the body circulation, such as the inferior vena cava and renal veins, either due to patent ductus venosus or abnormal shunt vessels [9]. The symptoms of CPSS range from asymptomatic to symptomatic such as hyperammonemia, abnormal liver function, manganese deposition in the brain, and pulmonary hypertension, depending on the shunt blood flow. Hypergalactosemia in neonatal period needs awareness as it is the first sign of CPSS. When these symptoms are observed in patients with polysplenia, CPSS should always be differentiated by aggressive abdominal imaging. The criteria for the indication of treatment are not yet established and should be considered on a case-by-case basis. There are two types of CPSS, intrahepatic and extrahepatic. Intrahepatic CPSS often closes spontaneously within 1–2 years of birth. If it does not close spontaneously, closure of the shunt vessel by laparotomy, laparoscopy, or catheterization (coil embolization) should be considered. In the case of CPSS with complete aplasia of the intrahepatic portal vein, liver transplantation is the only treatment, but recent advances in imaging have shown that complete portal vein aplasia is rare, and the indications for shunt closure are expanding.

### 1.8. Fontan Circulation in Heterotaxy Syndrome: Recent Concepts for Pulmonary Hypertensive Vascular Disease and Protein-Loosing Enteropathy

In heterotaxy syndrome with single ventricle physiology, which is a common complication, especially in right isomerism, Fontan-type operation enables separation of the systemic circulation from the pulmonary circulation and improves hypoxia. However, since the upper and lower vena cava and pulmonary arteries are anastomosed in this procedure, the pulmonary circulation is maintained by the difference between mean pulmonary artery pressure (mPAP) and mean left atrial pressure (transpulmonary pressure gradient; TPG), which corresponds to the driving pressure of the pulmonary circulation and the kinetic energy of systemic ventricular contraction. Therefore, if the pulmonary vascular resistance (PVR) increases even slightly, the pulmonary circulation does not flow easily, resulting in right heart failure due to venous congestion and left heart failure due to decreased preload. In these cases, pulmonary vasodilator therapy has recently been considered with the goal of maintaining a low PVR, although they do not meet the definition of pulmonary hypertension based on mPAP. Such condition represents the Panama classification category 3, a pediatric pulmonary hypertensive vascular disease secondary to cardiovascular disease [10].

Protein-loosing enteropathy (PLE) is a common postoperative complication of Fontan-type operations, occurring in 4%–13% of patients, especially those with heterotaxy syndrome, and once it occurs, it is often refractory and has a poor prognosis, with frequent blood transfusions and hospitalization reducing quality of life. However, the full extent of the disease remains to be elucidated and no absolute cure is established. Recently, Itkin et al. reported an improvement in cases of PLE after Fontan-type operation by embolization of a lymphatic fistula draining from the liver into the intestine [11]. The patients included eight cases (5 males and 3 females; aged 4–51 years; median 19.5 years) with underlying heterotaxy syndrome, central venous pressure 10–18 mmHg (median 14 mmHg), and duration of PLE 2 months–12 years (median 7.5 years). During the observation period of 84 to 1005 days (median 135 days) after lymphatic fistula embolization, only one of the eight patients remained unchanged, but the remaining seven patients showed improvement of PLE. We also started the same treatment very recently and obtained improvement of PLE in three out of three patients so far (T. Oyanagi, M. Inoue, H. Yamagishi, unpublished observation). In the Fontan circulation, the central venous pressure is higher than in the normal biventricular circulation, resulting in increased hepatic lymphatic flow and dilatation of the hepatoduodenal lymphatic vessels and anatomical disruption of the hepatoduodenal lymphatic barrier. As a result, protein-rich lymph leaks into the intestine. Since intrahepatic lymph fistula embolization improves PLE, a new concept from this treatment is that the PLE develops when this change in lymphatic anatomy coincides with an increase in central venous pressure. This concept is interesting from the viewpoint that it could explain the lack of correlation between the onset of PLE and the severity of elevated central venous pressure [11].

### 1.9. Development of the Outflow Tract Region of the Heart and Congenital Heart Disease

At around 28 days of fetal life, the primitive heart tube loops to reveal the morphology of the left and right ventricles, and the conotruncus in the form of a single conduit grows between the right ventricular primordium, also called the bulbus cordis and the aortic sac [1,2,3]. As the conotruncus grows longer, conotruncal cushions/swellings develop from the left and right sides, twisting and fusing to form the conotruncal septum in a spiral fashion. At the same time, the opening of the conotruncus into the right ventricle moves to the left. As a result of the series of processes, the pulmonary artery and aorta separate and align with the right and left ventricles, respectively. The conal septum and the membranous septum are joined to form the ventricular septum. The subpulmonic conus persists and the subaortic conus is absorbed, completing the correct alignment of the great vessels and ventricles.

Based on developmental and morphological studies, it has been postulated that abnormalities in each of the above steps of the outflow tract development process lead to the following mechanisms of congenital heart disease (Figure 3): (1) double outlet right ventricle (DORV): impaired leftward migration of conotruncus and/or abnormal persistence/absorption of subarterial conus, (2) persistent truncus arteriosus (PTA): insufficient formation of the conotruncal septum, (3) transposition of the great arteries (TGA): insufficient twisting of the conotruncal septum and/or abnormal persistence/absorption of subarterial conus, and (4) tetralogy of Fallot (TOF): hypoplasia of subpulmonic conus and/or malalignment of aorta and the left ventricle12).

### 1.10. Congenital Outflow Tract Defects: Recent Concepts from Developmental Cardiology

The etiology of congenital heart diseases classified as cardiac outflow tract defects is still largely unknown. The most frequent genetic abnormality is the 22q11.2 deletion, involving TBX1 [12]. The 22q11.2 deletion is frequently associated with PTA and TOF but less frequently with TGA, suggesting that the molecular mechanism of pathogenesis may not be common. In the case of nonsyndromic diseases, there are very few cases that can be explained by a single genetic cause, and it is assumed that most cases are due to multifactorial inheritance, including environmental factors.

A new concept of the pathogenesis of PTA and TOF was developed in light of the developmental and cellular interactions of two cardiac progenitor cells, namely second heart field (SHF) cells and cardiac neural crest (CNC) cells [12,13,14,15,16,17,18]. The pathogenesis of PTA is thought to be the result of a complete loss of the conotruncal septum due to a developmental abnormality of CNC cells that normally give rise to the conotruncal cushions/swellings. The pathogenesis of TOF is thought to be due to impaired alignment of the conotruncal septum and muscular septum, resulting in the aortic overriding on top of the ventricular septal defect. On the other hand, there is another theory that hypoplasia of the conus below the pulmonary valve is the cause of infundibular stenosis of the right ventricle and malalignment of the conotruncal septum. The former theory suggests that the pathogenesis of TOF is mainly due to abnormalities in CNC cell development, while it is mainly due to abnormalities in SHF cell development in the latter. Furthermore, if the developmental abnormality of SHF cells is so severe that the main pulmonary artery is not formed at all, the only outflow tract from the heart is the aorta, which is presumed as the morphology of TOF with pulmonary atresia or PTA. Genes involved in the development and cellular interactions of SHF and CNC may be candidates for disease cause.

As for TGA, few genetic candidates have been reported, mainly genes involved in the formation of the left–right axis of the body, rather than genes involved in the development and cell–cell interaction of SHF and CNC [1,19]. TGA is often associated with heterotaxy syndrome, suggesting that there may be a mechanism whereby only the alignment of the great arteries is disturbed by minor abnormalities in left–right axis-related genes rather than outflow tract-related genes. There is a wide spectrum of disease in the DORV, and the pathogenesis and genetic cause of DORV may differ between the TOF type and the TGA type. Although DORV is often associated not only with 22q11.2 deletion syndrome but also heterotaxy syndrome, the molecular mechanism of the disease is different in each type, and the genetic heterogeneity seems to be strong.

## 2. Conclusions

Clinical developmental cardiology is an exploration of the mysteries of nature and biology and should be developed as a basis for elucidating the etiology of congenital heart disease. In this paper, the elucidation of the developmental mechanism of the left–right axis of the body, or the discovery of “Nodal flow” and that of the outflow tract of the heart, or the identification of “Second heart field and TBX1”, in this century are reviewed. These insights have added new pages to “clinical developmental cardiology” and deepened our understanding for the pathogenesis of heterotaxy syndrome and congenital outflow tract defects. The summary of selected genes associated with heterotaxy and various CHD is shown in Table 1. Additional clinical developmental cardiology would bring us closer to the “essence of nature” and contribute to the advancement of medicine.

## Figures and Tables

**Figure 1 jcm-11-02381-f001:**
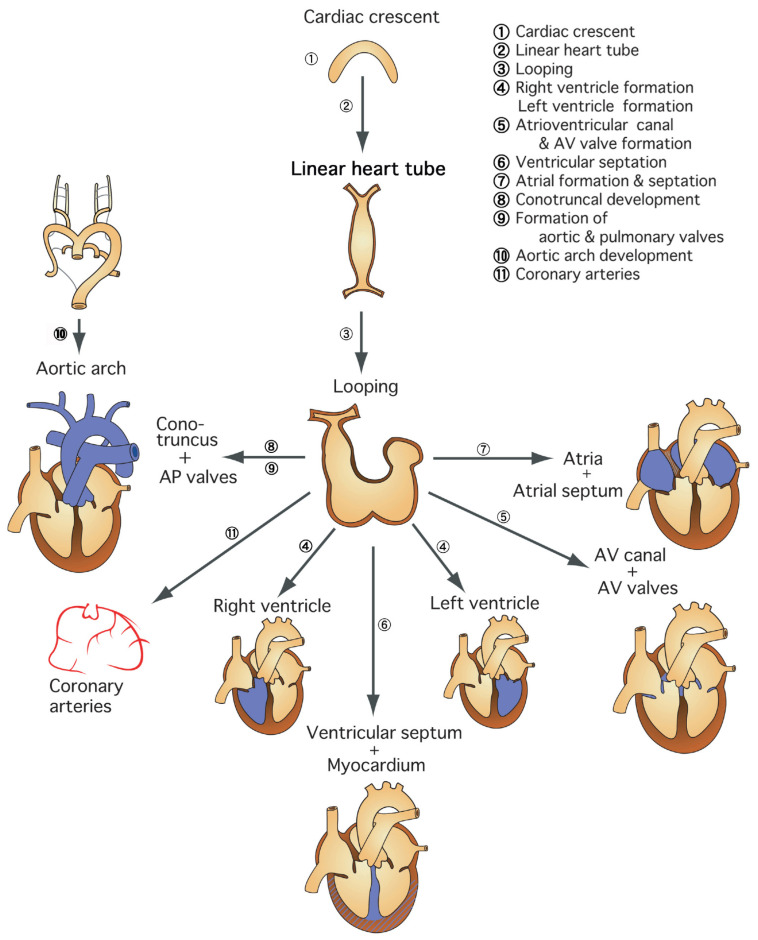
Molecular embryology for an understanding of the individual modular steps in cardiovascular development. To better understand this complex process, it may be helpful to divide it into several regions or steps. Because most congenital heart diseases that we encounter in our daily practice are specific developmental abnormalities in one of these regions, this concept is also important for understanding the etiology of congenital heart disease.

**Figure 2 jcm-11-02381-f002:**
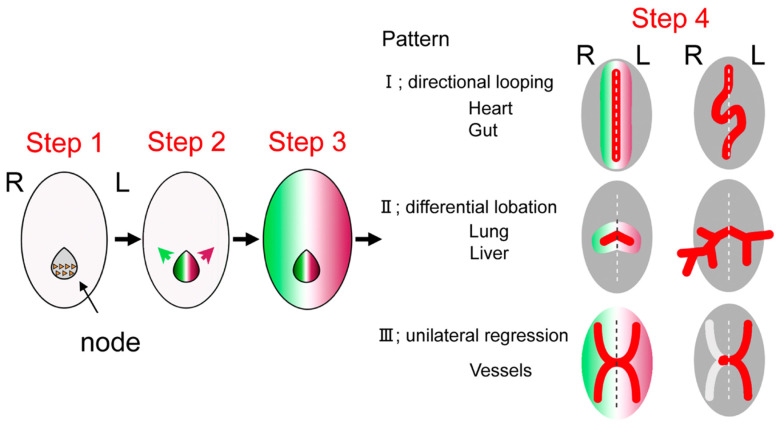
The four-step process of determining the left–right axis of the body including the heart. Step 1: rotational movement of cilia in the node, Step 2: right-to-left nodal flow of the embryo, Step 3: expression of the “left-sided formation mechanism” on the left side of the embryo, and Step 4: activation of left-sided formation molecules and genes (Nodal–Lefty–Pitx2) to pattern each organ in asymmetric fashion (adapted from [5] Yashiro K, Miyakawa S, Sawa Y (2017) Molecular Mechanism Underlying Heterotaxy and Cardiac Isomerism. *Pediatric Cardiology and Cardiac Surgery* 33, 349–361.).

**Figure 3 jcm-11-02381-f003:**
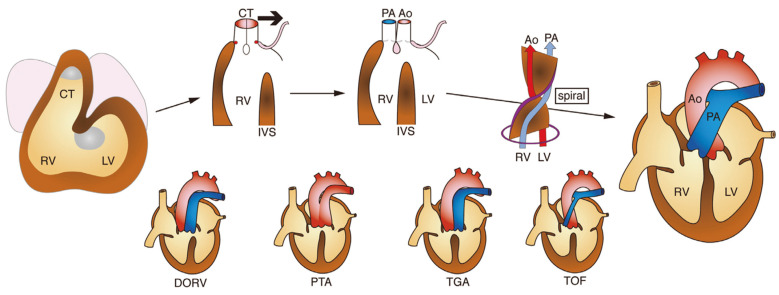
Normal development and congenital defects of the cardiac outflow tract. Based on developmental and morphological studies, abnormalities in each step of the outflow tract development (upper figures) lead to the spectrum of congenital heart disease involving the outflow tract (lower figures). Bold arrow represents leftward movement of conotruncus (CT). RV: right ventricle, LV: left ventricle, IVS: interventricular septum, Ao: aorta, PA: pulmonary artery, DORV: double outlet right ventricle, PTA: persistent truncus arteriosus, TGA: transposition of the great arteries, and TOF: tetralogy of Fallot.

**Table 1 jcm-11-02381-t001:** Selected genes associated with heterotaxy and various congenital heart disease (CHD). AS, aortic stenosis; ASD, atrial septal defect; AVSD, atrioventricular septal defect; BAV, bicuspid aortic valve; DORV, double-outlet right ventricle; HLHS, hypoplastic left heart syndrome; IAA, interruption of aortic arch; PTA, persistent truncus arteriosus; SVAS, supravalvular aortic stenosis; TOF, tetralogy of Fallot; VSD, ventricular septal defect.

Gene	Gene Function	Related CHD
*ZIC3*	Transcription factor	Heterotaxy
*NODAL*	TGF-β signal	Heterotaxy
*CFC1*	Nodal pathway	Heterotaxy
*ACVR2B*	Nodal pathway	Heterotaxy
*FOXH1*	Nodal pathway	Heterotaxy
*LEFTY A*	Nodal pathway	Heterotaxy
*GDF1*	Nodal pathway	Heterotaxy
*SESN1*	Nodal pathway	Heterotaxy
*CRELD1*	EGF-like protein	Heterotaxy
*DNAI1*	Dynein arm component	Heterotaxy
*DNAH5*	Dynein arm component	Heterotaxy
*NKX2-5*	Transcription factor	ASD, TOF, HLHS
*NKX2-6*	Transcription factor	PTA
*GATA4*	Transcription factor	ASD, AVSD, TOF
*GATA6*	Transcription factor	PTA, TOF
*TBX1*	Transcription factor	PTA, TOF, IAA
*TBX5*	Transcription factor	AVSD, ASD, VSD
*TBX20*	Transcription factor	ASD, VSD
*ZFPM2/FOG2*	Transcription factor	TOF, DORV
*NOTCH1*	Notch pathway	BAV, AS, TOF
*VEGFA*	Cell signaling	TOF, AS, IAA
*ELN*	Structural protein	SVAS
*MYH6*	Structural protein	ASD
*MYH7*	Structural protein	Ebstein’s anomaly

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
