# Peer review of "Clinical Developmental Cardiology for Understanding Etiology of Congenital Heart Disease"

_jcm, 2022, doi:10.3390/jcm11092381_

Round 1
Reviewer 1 Report
The review under evaluation deals with congenital heart disease and the embryological mechanisms that can bring about the changes that lead to this structural heart disease.
The determination of left-right patterning in our body proceeds through steps involving leftward 'nodal flow' by motile cilia in the node and molecules that are only expressed on the left side of the embryo, ultimately activating the molecular pathway for left-side specific morphogenesis. Disruption of any of these steps may lead to left-right patterning defects or heterotaxy syndrome.
This review teaches us that to better understand the etiology of CHD, it is important to consider which type of CHD is caused by abnormalities at each stage during the complex development of the cardiovascular system.
The article is well structured, the images make clear the various embryological stages and the pathologies that may result.
The use of English is correct and well understood.
I think it would be appropriate to add a summary table of the various existing congenital heart diseases and the genes involved, highlighting the mechanisms of mutation with the physically determined effects, to be able to better list the various CHD.
Author Response
I think it would be appropriate to add a summary table of the various existing congenital heart diseases and the genes involved, highlighting the mechanisms of mutation with the physically determined effects, to be able to better list the various CHD.
Response: Thank you very much for the review and the important suggestion. According to the suggestion, I add a summary table of various congenital heart diseases and the genes involved highlighting the gene function/mechanism as Table 1 in the conclusion section (“Concluding remarks").
Reviewer 2 Report
1) Page 4, line 128: write bacterial strains in italics: pneumoniae and H. influenzae
2) Update the bibliography since only 26% of the references correspond to the last ten years. Usually, in a review article, at least 51% of the references included should be in this range.
3) Improve the conclusion section.
Author Response
1) Page 4, line 128: write bacterial strains in italics: pneumoniae and H. influenzae
Response: Thank you very much for the correction. These were corrected in italics in the revised version.
2) Update the bibliography since only 26% of the references correspond to the last ten years. Usually, in a review article, at least 51% of the references included should be in this range.
Response: Thank you very much for the important point. According to the recommendation, the references were revised, resulting in 52.6% of them corresponds to the last ten years in the revised version.
3) Improve the conclusion section.
Response: Thank you very much for the suggestion. I improved the conclusion section with revising some sentences and adding a summary table (Table 1) in the revised version.
Round 2
Reviewer 2 Report
No one